# Influenza Virus Infections and Cellular Kinases

**DOI:** 10.3390/v11020171

**Published:** 2019-02-20

**Authors:** Robert Meineke, Guus F. Rimmelzwaan, Husni Elbahesh

**Affiliations:** Research Center for Emerging Infections and Zoonoses (RIZ), University of Veterinary Medicine (TiHo), Bünteweg 17, 30559 Hannover, Germany; robert.meineke@tiho-hannover.de

**Keywords:** influenza virus, kinases, antivirals, phosphorylation, small molecule inhibitors, metabolism, replication, pathogenesis

## Abstract

Influenza A viruses (IAVs) are a major cause of respiratory illness and are responsible for yearly epidemics associated with more than 500,000 annual deaths globally. Novel IAVs may cause pandemic outbreaks and zoonotic infections with, for example, highly pathogenic avian influenza virus (HPAIV) of the H5N1 and H7N9 subtypes, which pose a threat to public health. Treatment options are limited and emergence of strains resistant to antiviral drugs jeopardize this even further. Like all viruses, IAVs depend on host factors for every step of the virus replication cycle. Host kinases link multiple signaling pathways in respond to a myriad of stimuli, including viral infections. Their regulation of multiple response networks has justified actively targeting cellular kinases for anti-cancer therapies and immune modulators for decades. There is a growing volume of research highlighting the significant role of cellular kinases in regulating IAV infections. Their functional role is illustrated by the required phosphorylation of several IAV proteins necessary for replication and/or evasion/suppression of the innate immune response. Identified in the majority of host factor screens, functional studies further support the important role of kinases and their potential as host restriction factors. PKC, ERK, PI3K and FAK, to name a few, are kinases that regulate viral entry and replication. Additionally, kinases such as IKK, JNK and p38 MAPK are essential in mediating viral sensor signaling cascades that regulate expression of antiviral chemokines and cytokines. The feasibility of targeting kinases is steadily moving from bench to clinic and already-approved cancer drugs could potentially be repurposed for treatments of severe IAV infections. In this review, we will focus on the contribution of cellular kinases to IAV infections and their value as potential therapeutic targets.

## 1. Introduction

Influenza A (IAV) and B (IBV) viruses are important causes of upper respiratory tract infections [1]. IAV can cause severe acute respiratory disease with an attack rate of 5–10% in adults and 20–30% in children annually [2,3]. The significant public health burden caused by IAV infections is exemplified by the annual fatal cases globally, which number 290,000–650,000 [4]. Most at risk are children and the elderly, accounting for ~90% of case fatalities and/or complications [5,6]. Occasionally, novel antigenically distinct influenza A viruses emerge that may cause pandemic outbreaks as has occurred in 1918, 1957, 1968 and 2009. Unlike IAV, IBV viruses do not continuously circulate in animals and are therefore less likely to be associated with zoonotic transmission or pandemics [7]. However, they do co-circulate with IAV and can be significant contributors of influenza-related morbidity and mortality [7,8,9]. Vaccination is the preferred intervention against influenza viruses and helps to limit the impact influenza outbreaks may have. In addition, antiviral drugs are available for the treatment of influenza virus infections. The surface glycoprotein hemagglutinin (HA) is the major target for the induction of virus-neutralizing antibodies by vaccination. Currently available antiviral drugs against influenza inhibit the enzymatic activity of the viral neuraminidase (neuraminidase inhibitors (NAIs)), inhibit the viral M2 ion channels or inhibit viral RNA transcription by targeting components of the polymerase complex; all of which ultimately result in inhibition of virus replication [10,11]. The use of antiviral drugs may lead to the emergence of strains that have become resistant to these drugs by the accumulation of mutations greatly reducing their efficacy [12]. This, of course, is an important drawback and there is a need for treatment options that do not suffer from the emergence of resistant strains. Drugs that target host factors critical for virus replication may therefore be an attractive alternative.

IAVs are members of the *Orthomyxoviridae* family and have a negative-sense single-stranded RNA genome. Attachment of IAV to cell-surface receptors, containing either α2,3- or α2,6-linked sialic acid residues, initiates signaling cascades that facilitate internalization of the virus via receptor mediated endocytosis. During endosomal trafficking, pH-dependent fusion of viral and endosomal membranes leads to release of viral ribonucleoproteins (vRNPs) into the cellular cytoplasm where released vRNPs are shuttled to the nucleus for replication and transcription of viral RNA; all of which require host cell machinery [13]. These early events ultimately trigger multiple anti- and pro-viral pathways utilized, suppressed or evaded by IAV. The robust production of pro-inflammatory cytokines and chemokines observed during severe IAV infections is often referred to as a “cytokine-storm” (reviewed in [14]). This dysregulated immune response is associated with severe influenza induced pneumonia that can be fatal, especially in susceptible populations including children, older adults and the immunocompromised [15].

In contrast to IAV, IBV is understudied, with only a few studies addressing the role of host factors, and specifically kinases, and their role during IBV infections. A better understanding of the viral and cellular processes, mechanisms and interactions is required to develop new treatment options [7]. Considering the overlap of IAV- and IBV-utilized kinases and their related cellular signaling cascades to prime viral replication, defining these pathways is likely to help in developing comprehensive host-targeted antivirals against IAV and IBV.

Kinases link a myriad of external stimuli with downstream effectors through phosphorylation of proteins and/or lipids. So far, more than 500 kinases have been identified in the human kinome [16]. These kinases are typically categorized based on their phosphorylation substrate: tyrosine, serine/threonine or lipids; as well as kinases that have dual-specificity. Target residues (Tyr, Ser, Thr) are generally within well-defined consensus sequence motifs recognized by a given kinase [17,18,19]. Interestingly, the distribution of protein phosphorylation in eukaryotic cells is distributed at a ratio of ~1000:100:1 (serine:threonine:tyrosine) [20]. Phosphorylation can alter activity and subcellular localization, as well as biomolecular interactions [21]. In addition, phosphorylation can promote scaffolding activities of proteins that enhance, inhibit and modulate the substrates interaction with other cellular components [22]. Aberrant kinase activity is typically associated with several pathologies including cancer, diabetes or neurodegenerative diseases, which has led to the development and investigation of several kinase inhibitors for clinical use [23,24,25,26]. However, as of 2018, only 30 small-molecule kinase inhibitors (SMKIs) have gained FDA-approval for clinical use [26,27]. No SMKIs are currently under clinical trial investigation against influenza virus infections. IAV does not encode a kinase and is therefore dependent on cellular kinases to directly or indirectly, regulate phosphorylation-dependent processes including viral entry and uncoating, viral RNA and protein synthesis, protein relocation and release of viral particles [28,29,30,31]. In addition, several studies have illustrated the importance of IAV-protein phosphorylation in regulating viral replication and evasion/suppression of innate immune signaling cascades that control expression of pro inflammatory chemokines and cytokines response [32,33,34,35,36,37,38,39,40,41,42,43,44,45,46,47]. Moreover, RNAi screenings continue to add to the list of host factors that impact IAV replication [48].

Therefore, a better understanding is required of how influenza viruses utilize these critical host factors and how these factors regulate species-specific host adaption and pathogenesis of influenza viruses. This review aims to discuss current knowledge of the role cellular kinases play during in vitro and in vivo influenza virus infections as potential antiviral targets (Table 1) (Figure 1). Given the current state of knowledge, this review will be largely focused on IAV studies; however, IBV studies will be highlighted when possible.

## 2. Phosphorylation of Influenza Virus Proteins

Phosphorylation of IAV and IBV has been reported with some conservation across influenza virus species [34]. IAV protein-phosphorylation regulates different stages of the viral cycle by either promoting replication or evading/suppressing the innate immune response [32,33,34,35,36,69]. Moreover, treatment with kinase inhibitors affects influenza A virus RNA and protein synthesis, shuttling of viral proteins between the cytoplasm and nucleus, and virion release [28,29,30,31,80]. The nonstructural protein NS1 is a multifunctional immune modulator that counteracts host defenses [81,82]. NS1 phosphorylation at T215, S42 and S48 is thought to regulate the dsRNA binding capacity of NS1, which promotes evasion of the innate immune response [33,83]. Akt, an effector kinase of both the PI3K and ERK pathways, is responsible for T215 phosphorylation, which consequently results in viral entry and genome replication suppression following Akt inhibitor treatment [84,85]. Additionally, mutation of S42 eliminates the interaction of NS1 with dsRNA and attenuates viral replication [33,84]. T132 phosphorylation of the M1 protein controls its nuclear import, which is critical for viral replication. The Janus kinase 2 (JAK2) inhibitor AG490 prevents nuclear import of M1, suggesting that JAK2 might be responsible for M1 T132 phosphorylation [34,35]. Inhibition of IAV nucleoprotein (NP) phosphorylation leads to its nuclear retention that is largely regulated by several phosphorylation sites, including S9, Y10, S165 and Y296. Mutation of these sites results in decreased viral replication in vitro and in vivo largely through disruption of interactions with cellular importin-α and chromosomal maintenance 1 proteins [34,36,86].

## 3. Tyrosine Kinases

Tyrosine kinases (TK) are a subgroub of ~90 kinases within the human kinome that phosphorylate tyrosine amino acid residues; this can lead to conformational changes in a given protein or even serve as a scaffolding site to facilitate protein–protein interactions. TKs are further classified into receptor tyrosine kinases (RTK) and non-receptor tyrosine kinases (non-RTK). Non-RTK act as intracellular signal transducers, mediating the signaling of cell-surface receptors for cytokines, growth factors and other ligands [16]. Several phosphorylation sites (S, T and Y) in IAV and IBV proteins have been identified [34]. Based on additional sequence analysis of >50,000 strains (www.fludb.org), we identified highly conserved tyrosine residues in replication complex proteins (PA, PB1, PB2) and NP proteins of all IAV subtypes. This level of conservation suggests an evolutionary importance that might be exploited in understanding conserved functions and developing broadly active therapeutics targeting TK. Interestingly, while many of these phosphorylation sites have been previously reported and their importance demonstrated or inferred, the kinases that carry out their phosphorylation have yet to be experimentally validated.

Nerve growth factor receptor (TrkA) is a receptor tyrosine kinase that was shown to play a role in IAV viral RNA synthesis, vRNP nuclear export and virion release. In vitro inhibition of TrkA has been shown to diminish IAV RNA (vRNA, mRNA and cRNA) synthesis independently of NFκB signaling [29]. Interestingly, this reduction in RNA synthesis was largely due to direct inhibition of CRM1-mediated export and subsequent nuclear retention of IAV RNPs [29]. In addition, TrkA inhibition leads to reduced activation of the lipid biosynthesis enzyme, farnesyl diphosphate synthase (FPPS), which is known to modulate virion budding [28]. However, the exact mechanism of TrkA-mediated FPPS regulation remains undefined.

Focal adhesion kinase (FAK) is a non-RTK and a component of focal adhesions that tether the actin cytoskeleton to the extracellular matrix. We previously showed that FAK links phosphatidylinositol-3 kinase (PI3K) activation and cytoskeletal reorganization required for endosomal trafficking during IAV entry [40]. Furthermore, FAK positively regulates IAV replication and polymerase activity of different IAV strains/subtypes [39,40]. Others have also reported roles for FAK during other viral infections [87,88,89,90,91,92]. FAK can modulate the cellular immune response by regulating various functions of T cells, B cells and macrophages [93,94,95]. Consistent with this, we have observed FAK dependent regulation of innate immune responses during severe IAV infection in mice [96].

Abl1 (also known as Abelson murine leukemia viral oncogene homolog 1 or c-Abl) is a cytoplasmic and nuclear non-RTK that phosphorylates CRK (also known as p38 or proto-oncogene c-CRK), an adaptor protein required for efficient replication of avian influenza viruses and subsequent JNK-mediated apoptosis [97]. The viral nonstructural protein 1 (NS1) can disrupt Abl1-CRK interactions via its Src homology binding motifs and thereby inhibit CRK phosphorylation, ultimately resulting in IAV-subtype specific pathogenicity as shown for the 1918 pandemic H1N1 virus [50,51].

Acute respiratory distress syndrome (ARDS) and acute lung injury (ALI) due to immune cell infiltration during severe IAV and ensuing secondary bacterial infections can result in respiratory failure and are the main causes of death in influenza-infected patients [98]. Bruton’s tyrosine kinase (Btk) can regulate TLR4-mediated activation in human neutrophils [99]. Interestingly, chemical inhibition of Btk can alleviate IAV induced ARDS symptoms in mice [49]. This effect is likely due to limiting damaging neutrophil activity and production of pro-inflammatory chemokines and cytokines including TNF-α, IL-1β, IL-6, KC, and MCP-1 during acute lung injury [49].

The IFN receptor I and III-associated tyrosine kinase 2 (Tyk2) has emerged as an important host factor targeting secondary bacterial infections. The virally induced retention of IL-1β and GM-CSF diminishes the bacterial-induced innate immune response that may allow the establishment of secondary bacterial infection. Specific ex vivo inhibition of Tyk2 resulted in impaired bacterial growth due to restored IL-1β and GM-CSF levels in human alveolar tissues [52].

## 4. Serine/Threonine Kinases

Serine/Threonine kinases (STKs) facilitate phosphorylation of protein at either serine or threonine residues. STKs are central components of many cellular signaling pathways including Raf/MEK/ERK, nuclear factor kappa-B (NF-κB) and PKC [100,101,102]. The Ras-dependent Raf/MEK/ERK pathway is activated by almost all cytokines and growth factors that bind to receptor tyrosine kinases, cytokine receptors and G-protein coupled receptors [43]. Accordingly, the importance of Raf/MEK/ERK signaling for effective IAV replication has previously been demonstrated [31,60].

IAVs utilize multiple mechanisms to hijack STKs to evade subsequent innate immune responses. c-Jun N-terminal kinases 1 and 2 (JNK1/JNK2) can regulate pro-inflammatory response induction and are upregulated by several IAV strains. IAV-mediated induction of JNK1/JNK2 activity triggers the Raf/MEK/ERK pathway, mediating production of chemokines and cytokines including tumor necrosis factor alpha (TNF-α), interferon β (IFN-β) and interleukin 6 (IL-6) [45]. Interestingly, recent studies suggest that JNK1-dependent phosphorylation of Bcl-2, a process normally observed as a starvation induced autophagy signal, is promoted by viral JNK1 activation resulting in virus-induced autophagy [46]. Chemical inhibition of JNK1/JNK2 resulted in reduced levels of pro-inflammatory cytokines in vivo [45]. Additionally, in vitro inhibition of JNK1/JNK2 results in impaired vRNA synthesis; however, the mechanism is yet to be defined [53].

As a member of the mitogen-activated protein kinase (MAPK) family, p38 is involved in several steps of the IAV infection cycle. IAV infected cells expressing the antiapoptotic protein Bcl-2 show reduced viral titers due to reduced vRNP export from the nucleus with no effect on virally induced apoptosis. The antiapoptotic effect of Bcl-2 was reduced by phosphorylation of its threonine 56 and serine 87 residues by virus-induced p38 activity. Inhibition of p38 diminished viral replication, vRNP export and apoptosis [103]. During early stages, TLR4 mediated viral activation of p38 MAPK is important for viral entry and replication [37,55]. Furthermore, in vivo inhibition of p38 MAPK directly limited excessive cytokine expression through an IFN-dependent mechanism. This regulation is mediated via phosphorylation of STAT1 and subsequent engagement of the IFNβ promotor to regulate IFN-stimulated gene (ISG) expression [54]. Influenza virus induced perturbations of the intracellular redox balance resulting in increased production of reactive oxygen species (ROS) can also activate p38 [56,104]. Furthermore, NADPH oxidase 4 (NOX4)-regulated p38 and ERK activation leads to increased ROS production during IAV infections in vitro [56,105,106]. Interestingly, mouse experiments suggest that the effects of Bcl-2 and NOX4 may be gender dependent. Female mice exhibited reduced clinical symptoms and viral titers; in contrast, higher IAV replication in male mice correlated with higher expression of NOX4 and phosphorylation of p38 [107]. The NF-κB signaling pathway is a central regulator of innate immune responses and the IkB kinase (IKK) is a direct target of the viral NS1 protein in counteracting the NF-κB mediated cellular antiviral response [63,108]. However, the majority of publications have shown that inhibition of NF-κB signaling diminishes viral replication in vitro and in vivo [64,65,66]; more specifically, lowered levels of pro-inflammatory factors, reduced caspase activity and therefore impaired caspase-mediated nuclear export of vRNP [62]. Interleukin 1 receptor-associated kinase-M (IRAK-M) is a NF-κB signaling related cellular kinase. During IAV induced pneumonia, IRAK-M acts a central regulator of inflammation of mucosal tissue in the respiratory tract. IRAK-M knockout mice challenged with IAV showed strongly increased lethality rate and decreased viral clearance [67].

The successful nuclear export of vRNP has been shown to depend directly on the viral activation of the Raf/MEK/ERK signaling pathway [31,109]. MAPK kinase (MEK) and extracellular signal-regulated kinase (ERK), belong to the group of classical mitogen-activated protein kinases (MAPK). MEKs have been shown to regulate IAV and IBV replication [31,109]. Several MEK inhibitors resulted in vRNP retention, reduced titers of progeny virus in vitro, and also improved mouse survival in vivo [57,58,59]. During early stages of IAV infection, ERK regulates the vacuolar H^+^-ATPase (V-ATPase) activity to mediate pH-dependent acidification of endosomes and subsequent fusion of the viral and endosomal membranes [41]. In vitro inhibition of ERK, a direct downstream mediator of MEK, impedes IAV vRNP nuclear import as well as export [41,60]. IAVs activation of Raf/MEK/ERK signaling also induces p90 ribosomal s6 kinases (RSK), which play an important role as downstream mediators of ERK signaling [61,110]. RSK2 is involved in regulation of cell growth and proliferation. RSK2 knockdown using shRNAs results in increased IAV and IBV replication and IAV polymerase activity [61]. Inhibition of RSK2 blocked IAV-induced phosphorylation of double-stranded RNA-activated protein kinase (PKR), one of 4 known kinases (PKR, HRI, PERK and GCN2) that phosphorylate the translation-initiation factor elF2 during stress responses resulting in inhibition of cap-dependent translation of cellular and viral proteins [61,111]. PKR activation by influenza virus infections is well established and the virus has evolved multiple mechanisms to suppress PKR activation. Furthermore, IAV-dependent stimulation of NF-κB and IFN-β was impaired by RSK2 inhibition, suggesting an effect on the cellular antiviral response [61].

In addition to Raf/MEK/ERK kinases, the G protein-coupled receptor kinases (GRKs) are also implicated in the induction of innate immunity pathways. Recent phosphoproteomic studies identified GRK2 as an important junction of cellular signaling pathways activated by IAV. In vitro and in vivo inhibition of GRK2 resulted in decreased viral replication [71], while the exact function of GRK2 remains unclear. Polo-like kinases (PLK) act as GRK nodes of cellular signaling and are crucial regulators of cell division and the cell cycle [112]. PLK1 has been described as acting as a pro-viral host factor for several viruses by phospho-regulating viral proteins [113,114]. A recent study shows that in vitro and ex vivo inhibition, as well as knockdown of PLK1, PLK3 and PLK4, results in impaired IAV replication [73].

Protein kinase C (PKC) is a STK that regulates multiple cellular processes including proliferation, differentiation, apoptosis and angiogenesis. The functional versatility of PKC is dependent on its various isoforms responding to different stimuli. The complexity of eleven different PKC isoforms expressed in most tissues also limits understanding of their function within different cell types [115]. Nevertheless, Kurokawa et al. showed almost 30 years ago that general in vitro inhibition of PKC results in reduced viral protein synthesis [30]. More recent studies have further defined the function of PKC isoforms and their involvement in IAV infections. Treatment of cells with bisindolylmaleimide, a highly specific PKC inhibitor that has activity against most PKC isoforms, reversibly inhibits virus entry by blocking endosomal trafficking and virion uncoating of both IAV and IBV [80]. Phosphorylation of the viral proteins PB1 and NS1, important for polymerase activity and efficient viral replication, has been shown to be PKCα dependent in vitro [68] and for PB1 in vivo [69]. In PKCβII kinase-dead cells, IAV is retained in late endosomal compartments, suggesting PKCβII as an important modulator of IAV entry [44]. PKCδ, interaction with the IAV polymerase subunit PB2, regulates NP oligomerization and vRNP assembly, and ablation of PKCδ impaired replication of the viral genome in vitro [70].

## 5. Lipid Kinases

Lipid kinases are key mediators of intracellular signaling, central carbon and lipid metabolism, apoptosis and cell proliferation through phosphorylation of lipid residues. Several lipid kinases have been implicated in several steps of IAV replication and in modulating cellular antiviral responses [38,79,116,117,118]. One of the central lipid kinases is PI3K, which phosphorylates inositol phospholipids [119]. PI3K and its downstream effectors, Akt and mammalian target of rapamycin (mTOR), form a key signaling nexus that regulates cell differentiation, translation and metabolism [120]. Furthermore, it is involved in cross-interaction with other cellular signaling pathways including Raf/MEK/ERK and NF-κB pathways [121]. Early and late PI3K during IAV infections are key events required for IAV replication with distinct outcomes at different times of infection [38]. Early PI3K activity is triggered by viral attachment and mediates IAV entry [75]. Later during the infection, IAV NS1 suppresses PI3K activity via direct interactions with the p85 regulatory subunit. These interactions ultimately prevent AKT-mediated apoptosis, IRF-3 innate immune responses, vRNA synthesis and nuclear vRNP export [38,74,75,76,77,122,123]. It should be noted that IBV only minimally induces later PI3K activation or apoptosis. Furthermore, in contrast to IAV NS1, IBV NS1 is dispensable for the antiapoptotic effects of PI3K activation suggesting IBV has developed NS1-independent mechanisms to suppress apoptosis [116,124].

Sphingosin kinases (SphK1 and SphK2) are lipid kinases that control conversion of sphingosine to bioactive lipid sphingosine 1-phosphate (S1P) [125], a known modulator of Raf/MEK/ERK, NF-κB and PI3K/AKT/mTOR signaling pathways and regulator of apoptosis [126]. IAV upregulates SphK in in vitro infected cells influencing cellular signaling and promoting efficient influenza virus replication [78,79]. Chemical inhibition of SphK1 results in reduced vRNA synthesis via suppression of NF-κB activity and reduced vRNP nuclear export due to impaired activation of ERK and AKT [78]. SphK2 knockdown has also been shown to reduce IAV replication in vitro. Moreover, in vivo inhibition of SphK1 and SphK2 resulted in prolonged survival of mice challenged with IAV [79].

## 6. Linking Metabolism and Innate Immunity

Like many pathologic conditions, IAV infection alters the metabolic landscape and most of these alterations are mediated by kinases resulting in direct or indirect effect on IAV replication, infection kinetics and pathogenicity. Consistently, the majority of host-cell alterations following IAV infections are in metabolic pathways [127]. Virus regulated kinase activity can have a major influence on cellular metabolism. AMP-activated protein kinase (AMPK) is a major sensor and regulatory master switch of carbohydrate metabolism, and is directly involved in insulin signaling and lipid metabolism. It links central carbon metabolism and glucose availability with the host innate immune response [128,129,130,131]. AMPK activity is modulated by intracellular calcium levels and this activity can regulate the stimulator of interferon genes (STING) through UNC-51-like kinase 1 (ULK1) activation. STING serves as a crucial factor of the innate immune response and an essential mediator for recognition of intracellular bacterial and viral pathogens. STING-dependent IFNβ induction is regulated by the calcium-dependent membrane potential of mitochondrial membranes. In vitro inhibition of AMPK resulted in reduced TNF-α and IFN-β secretion after activation with the STING ligand 5,6-dimethyl xanthone-4 acetic acid (DMXAA) [132,133,134]. AMPK phosphorylation of multiple sites of ULK1 leads to its dissociation from AMPK and subsequent activation. ULK1 activity promotes phosphatidylinositol-3-phosphate (PI3P) synthesis that contributes to autophagosome formation in addition to JNK1 induced Bcl-2 dependent autophagy during IAV infection [46,135,136,137].

Although ER stress triggers translational shut-down through the PKR-like ER kinase (PERK), virally induced metabolic and ER stress in the context of an obese mouse model activates PKR [138]. This activation reduces cellular and viral translation and activates JNK1 and other inflammatory kinases in response [138]. Together, PKR and nutrient deprivation-dependent JNK1 activities lead to the subsequent activation of apoptosis signal regulating kinase 1 (ASK1) [139]. Integration of AMPK and JNK with other Raf/MEK/ERK related kinases allows engagement of metabolic processes via immune response components including NF-κB, PI3K/AKT/mTOR and PKC pathways [117,118,140,141,142]. Accordingly, the NF-κB regulating kinase, IKK, has recently been linked to glycolysis [143,144]. In addition, IKK- and PKC-dependent serine phosphorylation of the insulin receptor, inhibits insulin signaling and directly regulates cellular lipid metabolism [145,146]. Furthermore, PKC has been described to be involved in fatty acid fate regulation, auto-stimulating kinase activity [147]. PI3K/AKT/mTOR signaling mediates its effects upstream and downstream of NF-κB, Raf/MEK/ERK and PKC pathways to regulate lipogenesis and lipid metabolism [121,131,148,149]. Recent studies suggest that inhibition of Btk leads to metabolic stress through suppression of PI3K/AKT/mTOR signaling [150], highlighting the link between metabolism and innate immunity. Interestingly, using a PI3K/mTOR inhibitor to disrupt glucose metabolism in vitro results in reduced virus production independently of genome replication and most likely drives lipid membrane depletion due to viral budding [127]. It is important to note that influenza virus-induced kinase activity does not only serve to evade the immune response but can also promote a pro-viral metabolic environment and responses.

## 7. Perspectives and Future Directions

The continued threat of severe and potentially lethal influenza A virus outbreaks is highlighted by rapid viral evolution, emergence of novel subtypes and antiviral-resistant strains and limited vaccine efficacy. Developing virus-directed antivirals is akin to hitting a moving target. Therefore, approaches that largely mitigate the potential for drug-resistance while being effective against multiple IAV subtypes and strains is highly desirable. Therapies that target host cell factors meet these criteria and are more likely to avoid exuberant immune responses that are likely to reduce disease severity and improve patient outcome.

Kinases are ideal candidates for host-directed antiviral therapies by linking critical cellular processes utilized by most viruses. Moreover, their importance in pathologic conditions such as cancer has led to the development of small-molecule inhibitors and repurposing these clinically approved drugs to treat severe infectious diseases like influenza, should be exploited.

Several reports have recently highlighted critical roles for the focal adhesion kinase (FAK) pathway during infection by several viruses [87,88,89,90,91]. FAK is not only critical for embryonic development and expression of several cellular proteins, it also links integrins with actin reorganization and receptor endocytosis [151,152,153,154]. Given its role in several cancers and the unique structure of its kinase domain, FAK is an attractive target of anti-cancer therapies and several FAK inhibitors are under investigation for clinical use [155].

The FAK pathway has recently emerged as a nexus point engaging antiviral innate immune and inflammatory pathways. Accordingly, FAK is also a component of the intracellular RIG-I-like receptor antiviral pathway where it provides a link between perturbations of the cell surface receptor during viral entry and cytosolic innate immune sensors [156]. FAK modulates the cellular immune response by regulating T cells, B cells and macrophage functions [93,94,95]. FAK was also recently reported to directly phosphorylate IKKα thereby regulating canonical and non-canonical NF-κB pathways [157].

Although SMKIs have been met with often-warranted criticism, this has stemmed from a misconception in clinical literature and inaccurate distinction between in vitro/in vivo substrate (target) specificity and cell-population specificity in vivo of these SMKIs [158,159,160,161]. Because tyrosine kinases share conserved sequences in their ATP binding sites, ATP analogs have an increased likelihood of “off-target” effects on other kinases [162]. Therefore, new small molecule inhibitors designed to avoid this problem directly interfere with FAK autophosphorylation by binding to Y397 instead of blocking ATP binding. One such compound is FAK Inhibitor I (also known as Compound 14 or Y15) which has been validated as a selective FAK inhibitor [163,164,165]. We found that Y15-treatment of various cells, or expression of kinase-dead FAK mutant (FAK-KD), provided the first evidence that FAK is activated by IAV attachment and that FAK kinase activity is critical for efficient endosomal virus trafficking [40]. We also reported that inhibitor-treatment or FAK-KD expression reduced polymerase activity of multiple IAV subtypes including highly pathogenic H5N1 and H7N9. Importantly, we observed FAK interactions with the viral NP [39]; however, the significance of this interaction is still under investigation. Defactinib is an FDA approved FAK inhibitor that has dual activity against FAK and the related kinase Pyk2 and is therefore expected to have different effects than Y15 due to differences in specificities. Our published data utilizing Y15 clearly indicates a FAK specific role in IAV replication. However, given that Pyk2 has overlapping roles in immune cell development and functions [93,94,95], it is possible that inhibiting both kinases will have alternative outcomes. While this might first be viewed as a cause for concern, it provides the opportunity to potentially fine tune treatments where either FAK or Pyk2 or both can be inhibited depending on the timing of treatment (early vs late in infection).

Investigating repurposed cancer drugs for their antiviral properties and their potential immunomodulatory effects during infection will improve our understanding of the role of the respective kinases in the pathogenesis of IAV infections and may lead to the development of novel intervention strategies. Further research on the role of host kinases in virus-induced metabolic changes is warranted and will likely open-up additional avenues of basic and translational research.

## Figures and Tables

**Figure 1 viruses-11-00171-f001:**
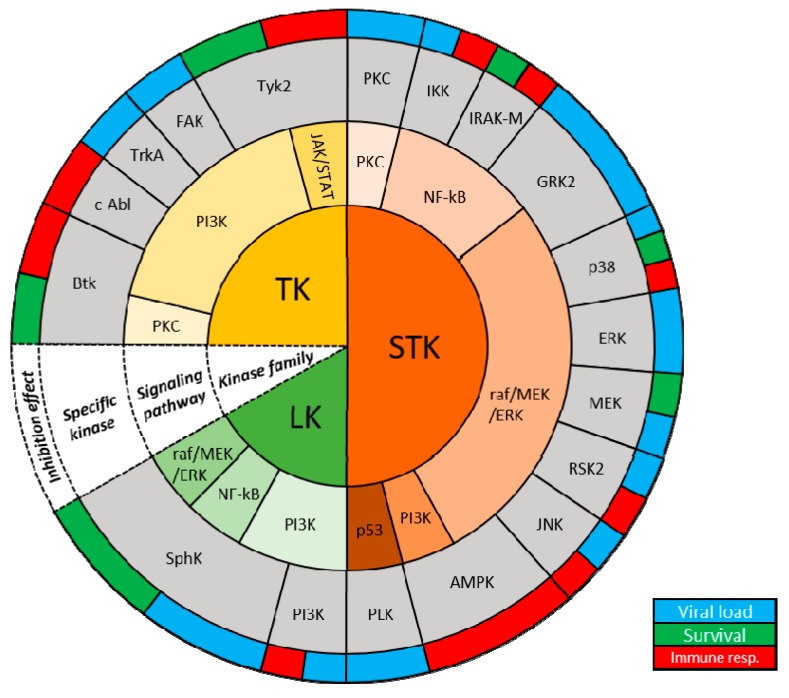
Host kinases and known roles during IAV infections. Schematic organizing host kinases based on kinase family, signaling pathway involved, specific kinase and effect of inhibition (from innermost to outermost ring; white cone).

**Table 1 viruses-11-00171-t001:** Overview of cellular kinases and their role in different stages of IAV replication.

	Name	IAV Effect	In Vitro, In Vivo or Ex Vivo	Inh. vs. KO	Reference
**Tyrosine**	FAK	-Virus entry -Polymerase activity	In vitro	Inhibition	Elbahesh et al., 2014, 2016 [39,40]
	TrkA	-vRNA synthesis -RNP export -Budding	In vitro	Inhibition	Kumar et al., 2011a, 2011b [28,29]
	Btk	-Neutrophil regulation	In vivo	Inhibition	Florence et al., 2018 [49]
	c-Abl	-Pathogenicity mediator	In vivo	Inhibition	Hrincius et al., 2014, 2015 [50,51]
	Tyk2	-Cytokine regulation	Ex vivo	Inhibition	Berg et al., 2017 [52]
**Serine/Threonine**	JNK1 / JNK2	-vRNA synthesis -Autophagy -Cytokine regulation	In vivo	Inhibition	Zhang et al., 2016, 2018; Xie et al., 2014 [45,46,53]
	P38 MAPK	-vRNA synthesis -RNP export -Prevents apoptosis -Cytokine regulation -Virus entry	In vivo	Inhibition	Borgeling et al., 2014; Choi et al., 2016; Marchant et al., 2010; Amatore et al., 2014 [37,54,55,56]
	MEK	-RNP export	In vivo	Inhibition	Haasbach et al., 2017, 2013; Droebner et al., 2011 [57,58,59]
	ERK	-RNP import -RNP export	In vivo	Inhibition	Pleschka et al., 2001, Marjuki et al., 2006 [31,60]
	RSK2	-Polymerase activity	In vitro	Knockdown	Kakugawa et al., 2009 [61]
	IKK	-Cytokine regulation -Caspase regulation -RNP export -Antiviral response modification	In vitro	Inhibition	Erhardt et al., 2013; Haasbach et al., 2013; Gao et al., 2012; Nimmerjahn et al., 2004; Wurzer et al., 2004 [62,63,64,65,66]
	IRAK-M	-Neutrophil interaction -Cytokine reg.	In vivo	KO	Seki et al., 2010 [67]
	PKC	-Endosomal entry -RNP assembly -Polymerase activity -Prevents apoptosis	In vivo	Inhibition	Mondal et al., 2017; Mitzner et al., 2009; Mahmoudian et al., 2009; Sieczkarski et al., 2003; Kurokawa et al., 1990 [30,44,68,69,70]
	GRK2	-viral uncoating	In vivo	Inhibition	Yanguez et al., 2018 [71]
	AMPK	-antiviral response	In vivo	Activation	Moseley et al., 2010 [72]
	PLK1/3/4	-unknown	ex vivo	KO	Pohl et al., 2017 [73]
**Lipid**	PI3K	-Virus entry -Prevents apoptosis -vRNA synthesis -RNP export -antiviral response modification	In vitro	Inhibition	Erhardt et al., 2006,2007; Shin et al., 2007; Erhardt and Ludwig, 2009; Ehrhardt, 2011; Marjuki et al., 2011 [38,41,74,75,76,77]
	SphK1 / SphK2	-vRNA synthesis -RNP export	In vivo	Inhibition	Xia et al., 2018; Seo et al., 2013 [78,79]

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
