# Peer review of "Influenza Virus Infections and Cellular Kinases"

_viruses, 2019, doi:10.3390/v11020171_

Round 1

Reviewer 1 Report

Meineke et al provide a concise but very informative review of the cellular kinase-controlled pathways involved during influenza virus infection. As the authors pointed out, current antiviral strategies against influenza are hampered by the virus' extraordinary ability to quickly mutate and become drug-resistant. The authors provide a nice overview of the importance of understanding cellular kinases in the context of their role during IAV infections and their potential as therapeutic targets. Intervention strategies that target the host, instead of the virus, to inhibit virus replication/pathogenesis is likely to be less affected by changes in the virus. 

In my modest opinion, the review is influenza type A centric and ignores the contribution of influenza type B as a major player in the morbidity and mortality associated with influenza virus infections. Although not as extensive and/or as studied as the type A viruses, the review could benefit if the authors discuss how much we know and do not know with respect to the role of cellular kinases during influenza B infections. Otherwise, the authors should clearly emphasize that the focus of the review is specifically on influenza A viruses.

In page 1, lines 25-26: In the sentence "..., and tyrosine kinases in particular, to IAV..." I read the review with equal emphasis on tyrosine kinases, STK and lipid kinases, so perhaps the authors should remove such mention in the sentence.

The first paragraph of the introduction needs a bit of work to have it better organized. Vaccination and the role of HA, antigenic drift and then antivirals and the effects of mutations on NA, M2 (which is not mentioned at all) and the newly approved drugs that target the polymerase complex (not mentioned either).

Page 4, Abl1 and CRK acronyms need to be spelled out. 

Page 5, line 108-109: Please indicate the subtypes or are these different pathotypes?

Page 5, lines 110-112: Most people that died of influenza-related complications, such as secondary bacterial pneumonia. The sentence reads as if most people die of influenza infection per se, which I think is only a minor proportion of the cases. Young infants are particularly vulnerable to flu B.

The perspectives and future directions should be expanded abut more to discuss how many drugs with activity against cellular kinases are actually in clinical trials to treat influenza and/or which are the most promising candidates. As written this section basically repeats parts of the abstract and the introduction. In my opinion, the authors should take advantage of the great literature review and provide their take on a pathway to mainstream use of cellular kinases inhibitors to treat/prevent influenza infections.

Finally, a figure summarizing  the pathways and viral genes (and SPNs) associated with protein-protein interactions should be really useful to the avid reader.

Author Response

Responses to Reviewer Comments: Virses-430925 (Meineke et al.)

We thank the Editor and Reviewers for their excellent and helpful comments. We appreciate the many constructive suggestions by the reviewers that helped us expand and improve our manuscript. We have included substantial revisions to address the reviewers’ suggestions and provide detailed responses below describing the changes.

REVIEWER 1

Comments and Suggestions for Authors

Meineke et al provide a concise but very informative review of the cellular kinase-controlled pathways involved during influenza virus infection. As the authors pointed out, current antiviral strategies against influenza are hampered by the virus' extraordinary ability to quickly mutate and become drug-resistant. The authors provide a nice overview of the importance of understanding cellular kinases in the context of their role during IAV infections and their potential as therapeutic targets. Intervention strategies that target the host, instead of the virus, to inhibit virus replication/pathogenesis is likely to be less affected by changes in the virus. 

In my modest opinion, the review is influenza type A centric and ignores the contribution of influenza type B as a major player in the morbidity and mortality associated with influenza virus infections. Although not as extensive and/or as studied as the type A viruses, the review could benefit if the authors discuss how much we know and do not know with respect to the role of cellular kinases during influenza B infections. Otherwise, the authors should clearly emphasize that the focus of the review is specifically on influenza A viruses.

Response 1: We have included a paragraph on IBV in the introduction to emphasize the importance of IBV. We have also included statements regarding the fact that in comparison to IAV, IBV is clearly understudied. Finally, we included results from studies describing IBV in addition to the IAV studies for several given kinases.

In page 1, lines 25-26: In the sentence "..., and tyrosine kinases in particular, to IAV..." I read the review with equal emphasis on tyrosine kinases, STK and lipid kinases, so perhaps the authors should remove such mention in the sentence.

Response 2: The sentence “...,and tyrosine kinases in particular,” was deleted. This review is indeed meant to highlight all kinase families without intentional bias.

The first paragraph of the introduction needs a bit of work to have it better organized. Vaccination and the role of HA, antigenic drift and then antivirals and the effects of mutations on NA, M2 (which is not mentioned at all) and the newly approved drugs that target the polymerase complex (not mentioned either).

Response 3: This paragraph has been revised as suggested. We have included M2 and polymerase inhibitors with appropriate citations. The effect or potential of antiviral resistance was mentioned and referenced in the original submission.

Page 4, Abl1 and CRK acronyms need to be spelled out. 

Response 4: Abelson murine leukemia viral oncogene homolog 1 or c-Abl (Abl1) and p38 or proto-oncogene c-CRK (CRK) are now spelled out to define the acronyms Abl1 and CRK in the text.

Page 5, line 108-109: Please indicate the subtypes or are these different pathotypes?

Response 5: The reference for this paragraph is on the NS1 interaction of 1918 H1N1 with Abl1. This is now clarified in the text to indicate the strain/subtype.

Page 5, lines 110-112: Most people that died of influenza-related complications, such as secondary bacterial pneumonia. The sentence reads as if most people die of influenza infection per se, which I think is only a minor proportion of the cases. Young infants are particularly vulnerable to flu B.

Response 6: The importance of secondary bacterial infections was further emphasized in this paragraph. The majority of influenza-related deaths are from ARDS and ALI. The impact of secondary bacterial infections during influenza virus infections on the development of these pathologies has been incorporated.

We have included the statement “...can be significant contributors of influenza-related morbidity and mortality” and included appropriate references.

As the focus of this review is the role kinases play during influenza virus infection, we limited the comparison of contribution to mortality/morbidity by IAV and IBV. However, the burden of disease by IBV on children is not easily assessed, as this seems to vary from season to season and is greatly influenced by the availability of matched vaccines. According to CDC and EuCDC, the 2017/2018 season is one of the few seasons that IBV was the major contributor to hospitalizations; it’s also interesting that during this season quadrivalent vaccine was not widely available in all countries and there was a mismatch with the trivalent vaccine.

The perspectives and future directions should be expanded abut more to discuss how many drugs with activity against cellular kinases are actually in clinical trials to treat influenza and/or which are the most promising candidates. As written this section basically repeats parts of the abstract and the introduction. In my opinion, the authors should take advantage of the great literature review and provide their take on a pathway to mainstream use of cellular kinases inhibitors to treat/prevent influenza infections.

Response 7: As suggested by the reviewer, we have significantly expanded this section while focusing on one signaling pathway (FAK) that is of significant interest in the field of cancer biology and is actively pursued for clinical trials. This pathway has emerged as an important pathway for influenza viruses as well as many other viruses.

We also included statements in this review underscoring the fact that there are no current clinical trials using any of FDA-approved or experimental kinase inhibitors to treat influenza virus infections. Moreover, there are only a limited number of studies where non-FDA approved SMKIs (still under clinical investigation) are used against other viruses (MERS and HIV).

Finally, a figure summarizing the pathways and viral genes (and SPNs) associated with protein-protein interactions should be really useful to the avid reader.

Response 8: We have generated a figure that summarizes that contribution of individual kinases to signaling pathways important during IAV infections and broadly highlighted the phenotype  resulting from inhibition of the given kinase. The mechanism by which the kinases are activated/suppressed and the viral components responsible is not well established outside of the well known roles of NS1. We have chosen not to include the viral genes in this figure as not to lend a false impression of their contribution and maintain focus on the downstream effect of the kinases involved. 

Reviewer 2 Report

The paper is aimed at reviewing the literature concerning the regulation of kinases mediated by influenza viruses and summarize the recent research about the inhibition of kinases as promising antiviral drugs. This is an interesting issue, but it should be deeply developed in some points and could be suitable for publication only after a careful revision of the text, according to following points:

1) Tyrosine kinases paragraph: As for other kinases, the authors should insert a brief introduction on of tyrosine kinases before describing different examples of each ones.

In the first sentence (lines 84-87) the identification of tyrosine residues in replication complex  proteins and NP in all IAV subtypes is reported. This concept should be better linked with the  rest of the paragraph, describing the potential role of these residues in viral replication.

References have to be updated: there is a recent paper describing another kinase TYK2 as an  important target for the immune regulation in case of severe influenza and secondary bacterial  pneumonia. Moreover, there is more recent literature that correlates the suppression of JNK1  mediated by influenza A virus and autophagy.

2)     Serine/Threonine Kinases paragraph: the literature concerning p38 lacks some important  findings and should be integrated. In particular, it has been demonstrated that p38 activity on the  influenza A virus cycle depends on Bcl-2 expression. Moreover, the mechanism through which  influenza A virus activates p38 through NOX4derived ROS production has been highlighted, as  well as the role of p38 phosphorylation in determining sex disparities in influenza virus infection. Finally, there is literature about the use of natural or synthetic compounds with antiinfluenza  efficacy mediated by the suppression of IAV-induced activations of p38, JNK and NF-kB pathways.

3)     Lipid Kinases paragraph: As for tyrosine kinases, before describing the correlation with  influenza virus infection a brief introduction is recommended.

4)     Linking metabolism and innate immunity: The link between immune response to IAV and  changes in metabolism status mediated by kinases should be deeply discussed. Literature about the association between AMPK and STING signaling modulation in innate immune response to  viruses should be mentioned. Recent progresses have been made to define the various pathways and kinases involved in the control immune response and should be managed in the text (e.g. metabolic stress and the immune system).

5)     The number of each reference should be inserted in the table so that the reader can easily  found it in the text.

Author Response

REVIEWER 2

 Responses to Reviewer Comments: Virses-430925 (Meineke et al.)

We thank the Editor and Reviewers for their excellent and helpful comments. We appreciate the many constructive suggestions by the reviewers that helped us expand and improve our manuscript. We have included substantial revisions to address the reviewers’ suggestions and provide detailed responses below describing the changes.

Comments and Suggestions for Authors

The paper is aimed at reviewing the literature concerning the regulation of kinases mediated by influenza viruses and summarize the recent research about the inhibition of kinases as promising antiviral drugs. This is an interesting issue, but it should be deeply developed in some points and could be suitable for publication only after a careful revision of the text, according to following points:

1)   Tyrosine kinases paragraph: As for other kinases, the authors should insert a brief introduction on of tyrosine kinases before describing different examples of each ones.

Response 1:  An introduction on tyrosine kinases is now added to the paragraph including information on substrate specificity, their proportion in the human kinome, different types of tyrosine kinases and their role for cellular signaling.

In the first sentence (lines 84-87) the identification of tyrosine residues in replication complex  proteins and NP in all IAV subtypes is reported. This concept should be better linked with the  rest of the paragraph, describing the potential role of these residues in viral replication.

Response 2: We have included a section entitled “Phosphorylation of influenza virus proteins” to highlight the known/predicted phosphorylation sites within the influenza virus genome and the potential kinases involved. We have also better defined the known/reported functions of some of these phosphorylations in both IAV and IBV.

References have to be updated: there is a recent paper describing another kinase TYK2 as an  important target for the immune regulation in case of severe influenza and secondary bacterial  pneumonia. Moreover, there is more recent literature that correlates the suppression of JNK1  mediated by influenza A virus and autophagy.

Response 3: We added the paper on the role of Tyk2 by Berg et al (2017) to the tyrosine kinases section. To our knowledge, the JNK1 data by Zhang et al. (2018) reference is the most recent published data.

2)     Serine/Threonine Kinases paragraph: the literature concerning p38 lacks some important  findings and should be integrated. In particular, it has been demonstrated that p38 activity on the influenza A virus cycle depends on Bcl-2 expression. Moreover, the mechanism through which  influenza A virus activates p38 through NOX4 derived ROS production has been highlighted, as  well as the role of p38 phosphorylation in determining sex disparities in influenza virus infection. Finally, there is literature about the use of natural or synthetic compounds with antiinfluenza  efficacy mediated by the suppression of IAV-induced activations of p38, JNK and NF-kB pathways.

Response 4: The suggested references on Bcl-2 dependent p38 activity were added to the paragraph describing the anti-apoptotic effect of Bcl-2, the inhibition of the effect through virus induced p38 activity and the resulting phosphorylation of Bcl-2 by p38 (Nencioni et al. (2009). Furthermore, the IAV induced increase in NOX4-dependend ROS and gender dependency has been added.

This review is focused on the role of kinases in viral replication, as such, we have minimized discussion of therapeutics (FDA-approved, natural, synthetic or otherwise) which we believe would be outside the scope of the review. The provided description of inhibitor effects is only meant to highlight the function/importance of kinases/pathways.

3)     Lipid Kinases paragraph: As for tyrosine kinases, before describing the correlation with  influenza virus infection a brief introduction is recommended.

Response 5: As suggested, an introduction on lipid kinases is now added to the paragraph including information on substrate specificity and their role for cellular signaling.

4)     Linking metabolism and innate immunity: The link between immune response to IAV and  changes in metabolism status mediated by kinases should be deeply discussed. Literature about the association between AMPK and STING signaling modulation in innate immune response to  viruses should be mentioned. Recent progresses have been made to define the various pathways and kinases involved in the control immune response and should be managed in the text (e.g. metabolic stress and the immune system).

Response 6: We have significantly revised this section to include the suggested literature on AMPK/STING, ULK1 and Bcl-2. We have also included additional information on stress/immunity in this section to highlight the interplay between the two systems.

5)     The number of each reference should be inserted in the table so that the reader can easily  found it in the text.

Response 7: Reference numbers have been inserted to the table as suggested.

Reviewer 3 Report

The review by Meineke et al provides a comprehensive collection of papers dealing with interplay of IAV and host cell kinases with a brief comment regarding  the central findings of each study. While quite useful as such, the review lacks perspective and opinions that make professionally and authoratively written reviews a pleasure to read. This review is only somewhat more informative than browsing through the titles of papers coming up in a PubMed search with keywords "IAV" and "kinase". 

Specific comments:

A chapter summarizing the results obtained in cell culture and in animal models using various FDA approved and other kinase inhibitors as experimental IAV therapy would strengthen this review.

In the abstract the authors say that kinases have been "identified as restriction factors in the majority of host factor screens".  I guess some kinases could really act as restriction factors for IAV in the correct virological meaning of this term, but believe that the authors here incorrectly use "restriction factor" to mean "hits" or "critical host cell co-factors" or something like this. 

Author Response

 Responses to Reviewer Comments: Virses-430925 (Meineke et al.)

We thank the Editor and Reviewers for their excellent and helpful comments. We appreciate the many constructive suggestions by the reviewers that helped us expand and improve our manuscript. We have included substantial revisions to address the reviewers’ suggestions and provide detailed responses below describing the changes.

REVIEWER 3

Comments and Suggestions for Authors

The review by Meineke et al provides a comprehensive collection of papers dealing with interplay of IAV and host cell kinases with a brief comment regarding  the central findings of each study. While quite useful as such, the review lacks perspective and opinions that make professionally and authoratively written reviews a pleasure to read. This review is only somewhat more informative than browsing through the titles of papers coming up in a PubMed search with keywords "IAV" and "kinase". 

We strongly disagree with the reviewer’s assessment in the contribution this review makes to an already understudied field. While we agree that we do provide a “comprehensive” view of studies that describe IAV-kinase interactions and providing their central findings, we also avoid the biases associated with “opinions”.

As suggested by the other reviewer, we have significantly expanded this section to focus on one signaling pathway (FAK) that is of significant interest in the field of cancer biology and is actively pursued for clinical trials. This pathway has emerged as an important pathway for influenza viruses as well as many other viruses. We believe this is a more appropriate section for opinions and perspectives.

Specific comments:

A chapter summarizing the results obtained in cell culture and in animal models using various FDA approved and other kinase inhibitors as experimental IAV therapy would strengthen this review.

Response 1: This review is focused on the role of kinases in viral replication; therefore, we have minimized discussion of therapeutics (FDA-approved, natural, synthetic or otherwise) which we believe would be outside the scope of the review. The provided description of inhibitor effects is only meant to highlight the function/importance of kinases/pathways.

It should also be noted that are no current clinical trials using any FDA-approved or experimental kinase inhibitors to treat influenza virus infections. Moreover, there are only a limited number of studies where non-FDA approved SMKIs (still under clinical investigation) are used against other viruses (MERS and HIV).

In the abstract the authors say that kinases have been "identified as restriction factors in the majority of host factor screens".  I guess some kinases could really act as restriction factors for IAV in the correct virological meaning of this term, but believe that the authors here incorrectly use "restriction factor" to mean "hits" or "critical host cell co-factors" or something like this. 

Response 2: We have changed the sentence in the abstract to “Identified in the majority of host factor screens, functional studies further support the important role of kinases and their potential as host restriction factors.”

In this regard, we have chosen a less-narrow definition of “restriction factors” that takes into account the indispensable role of host factors. Viruses do not replicate in a vacuum and their dependence on host factors for every step of the replication cycle makes them vulnerable to host factor availability, tissue expression and evolutionary conservation. As such, the absence (by genetic variation, therapeutic interventions, etc) of these “critical host factors” can restrict virus replication. It is therefore difficult to understand how some of these kinases would not be restriction factors.

Round 2

Reviewer 1 Report

The authors have made the necessary improvements and have properly addressed all my previous comments. Nothing more to add, a very nice review!

Author Response

REVIEWER 1

Comments and Suggestions for Authors

The authors have made the necessary improvements and have properly addressed all my previous comments. Nothing more to add, a very nice review!

Response 1: No response required

Reviewer 3 Report

The revised manuscript is now better than the original one, but suffers from the same limitations listed in my original comments.

Since the last two concluding sentences of the Abstract read "The feasibility of targeting kinases is steadily moving from bench to clinic as already approved cancer drugs are repurposed for treatments of severe IAV infections. In this review, we will focus on the contribution of cellular kinases to IAV infections and their potential as therapeutic targets" the reader would expect this aspect to be better discussed in thus review. When suggesting to discuss "experimental IAV therapies" using kinase inhibitors I obviously did not mean clinical trials but any relevant in vitro or animal models that could shed light into this question. If the authors are unwilling to extend their review to this direction, they should at least modify the abstract in order not to mislead and disappoint a potential reader.

The revised sentence on host cell kinases as restriction factors is no better than the original one. The authors use the term "restriction factor" in the exactly opposite meaning than the rest of the virology community  ("a less-narrow definition" indeed!) NB, restriction factors are effector molecules of intrinsic immunity, i.e. cellular antiviral factors that have to be missing or inactivated by viral proteins in order for efficient viral replication to occur.

Author Response

REVIEWER 3

Comments and Suggestions for Authors

The revised manuscript is now better than the original one, but suffers from the same limitations listed in my original comments.

Since the last two concluding sentences of the Abstract read "The feasibility of targeting kinases is steadily moving from bench to clinic as already approved cancer drugs are repurposed for treatments of severe IAV infections. In this review, we will focus on the contribution of cellular kinases to IAV infections and their potential as therapeutic targets" the reader would expect this aspect to be better discussed in thus review.

Response 1: This sentence has been toned-down to state “The feasibility of targeting kinases is steadily moving from bench to clinic and already-approved cancer drugs could potentially be repurposed for treatments of severe IAV infections.”

Furthermore, we modified the wording in the last sentence of the abstract “...and their value as potential therapeutic targets.” to avoid misleading the readership.

When suggesting to discuss "experimental IAV therapies" using kinase inhibitors I obviously did not mean clinical trials but any relevant in vitro or animal models that could shed light into this question. If the authors are unwilling to extend their review to this direction, they should at least modify the abstract in order not to mislead and disappoint a potential reader.

Response 2: This comment is somewhat puzzling as we provided in the original submission a table that indicates specific kinases, their effect on specific steps of virus infection, model (in vitro, in vivo or ex vivo), type of disruption and relevant references.

This was further elaborated in the revision by including an additional figure that shows the “phenotype” associated with disruption of specific kinases.

We respectfully believe that this has addressed the reviewer’s concern without identifying/highlighting specific inhibitors (e.g. Y15, BAY117082, 203580, etc) which may bias readers to using one molecule over another less well established but potentially more efficacious compounds.

The revised sentence on host cell kinases as restriction factors is no better than the original one. The authors use the term "restriction factor" in the exactly opposite meaning than the rest of the virology community  ("a less-narrow definition" indeed!) NB, restriction factors are effector molecules of intrinsic immunity, i.e. cellular antiviral factors that have to be missing or inactivated by viral proteins in order for efficient viral replication to occur.

Response 3: The original sentence states: “Identified as restriction factors in the majority of host factor screens, functional studies further support the important role of kinases.” This was changed in the revised manuscript to state: “Identified in the majority of host factor screens, functional studies further support the important role of kinases and their potential as host restriction factors.   (This revision was not “highlighted” in the revised manuscript)

It is not clear why our statement is inaccurate. Based on the reviewer’s assertion that “restriction factors are effector molecules of intrinsic immunity, i.e. cellular antiviral factors that have to be missing or inactivated by viral proteins in order for efficient viral replication to occur” Several kinases can fit this classical/dogmatic description including PKR, IKK, JAK and FAK all of which have been identified to exert antiviral activities through activation of IFN or NFkB pathways and are all either directly or indirectly inactivated or hijacked by the virus (references provided in manuscript). Furthermore, new restriction factors are emerging, kinases or otherwise, that do not fit this classical definition.

For example, the NS1 protein has been shown to alter the activity of several kinases (including PI3K and c-Abl) and this is required for adaptation of avian viruses to mammalian hosts. IAV protein phosphorylation by host kinases is critical for viral replication therefore IAV must adapt to utilize these kinases otherwise they are restricted.

We provided ample evidence where inhibition of specific kinases limits virus replication (Table-1). In defining “restriction factors”, whether the virus suppress or activates is irrelevant so long as this viral-induced activation/suppression leads to more efficient viral replication. This is precisely what we have described in our review and believe that we have not “overstepped” in our definition of restriction factors.